# Revealing the Dynamic Characteristics of Composite Material-Based Miura-Origami Tube

**DOI:** 10.3390/ma14216374

**Published:** 2021-10-25

**Authors:** Houyao Zhu, Zhixin Li, Ruikun Wang, Shouyan Chen, Chunliang Zhang, Fangyi Li

**Affiliations:** School of Mechanical and Electrical Engineering, Guangzhou University, Guangzhou 510006, China; houyaozhu@gzhu.edu.cn (H.Z.); 2111907011@e.gzhu.edu.cn (Z.L.); wrk@gzhu.edu.cn (R.W.); maxcsy@gzhu.edu.cn (S.C.)

**Keywords:** Miura tube, carbon fiber/epoxy resin, dynamic characteristics

## Abstract

Although Miura origami has excellent planar expansion characteristics and good mechanical properties, its congenital flaws, e.g., open sections leading to weak out-of-plane stiffness and constituting the homogenization of the material, and resulting in limited design freedom, should also be taken seriously. Herein, two identical Miura sheets, made of carbon fiber/epoxy resin composite, were bonded to form a tubular structure with closed sections, i.e., an origami tube. Subsequently, the dynamic performances, including the nature frequency and the dynamic displacement response, of the designed origami tubes were extensively investigated through numerical simulations. The outcomes revealed that the natural frequency and corresponding dynamic displacement response of the structure can be adjusted in a larger range by varying the geometric and material parameters, which is realized by combining origami techniques and the composite structures’ characteristics. This work can provide new ideas for the design of light-weight and high-mechanical-performance structures.

## 1. Introduction

Origami structures have attracted increasing attention in recent years, because of their numerous configurations and unique mechanical properties, e.g., negative Poisson’s ratio [1], bi-stability [2], and excellent energy absorption [3], to name a few. The goal of origami design is to design a specific crease patterns and then transform a sheet-like planar material into an exquisite three-dimensional structure by folding the material along these predefined creases [4]. Owing to the various benefits, including flexible design, simple manufacturing, and light weight, origami structures have demonstrated tremendous application potential in actual engineering for diverse fields, e.g., spacecraft solar panels [5,6], re-configurable structure design [7,8], energy-absorbing structures [9,10,11], biomedical equipment [12,13], foldable lithium-ion batteries [14,15], origami springs [16,17], origami robots [18,19,20], and sound barriers [21,22,23].

Among the various origami structures, Miura-ori is one of the most popular structures, which was invented by Miura in 1985 [5]. The Miura-ori consists of periodic arrays of unit cells in two directions on the plane, with the unit cells generally consisting of the same four parallelograms [5]. The unique mechanical properties of Miura-ori, which have been extensively studied in recent years, mainly come from the fruitful geometric topologies realized by folding, instead of the base materials. Some representative works are as follows: Lv [24] and co-workers studied the mechanical properties of periodically-arranged Miura sheets and found that the Miura sheet model can achieve both positive and negative Poisson ratios, which is consistent with its shear behavior and infinite bulk elastic modulus; in this study, a Miura sheet was considered to be a rigid origami structure, i.e., during the folding process, the facets between creases were not allowed to deform. Since then, more and more researchers have studied the stiffness characteristics of non-periodic origami structures and their load-bearing capacity, demonstrating their exotic mechanical properties [25,26,27,28,29,30,31,32]. Liu et al. [25] numerically and experimentally studied the deformation laws and energy-absorbing capacity of Miura sheets made of polymers under different loading conditions; the outcomes revealed that the dynamic properties of the Miura sheet can be broadly tailored. Xiang et al. [26] conducted experimental and numerical studies on the compression behavior of a Miura sheet made of nylon material, under quasi-static loading and impact loading; the relationships between the energy-absorbing capacity of Miura sheets and the unit’s acute angle were extensively investigated. Fischer and co-workers [27] numerically and analytically studied the buckling and crushing behavior of sandwich structures with Miura origami cores, revealing their excellent energy absorption. Sareh and Guest [28] creatively designed a class of non-isomorphic symmetric Miura-ori derivatives, including globally planar, globally curved, and flat-foldable tessellations. Most recently, quasi-static in-plane compression of novel metamaterials, inspired by zig-zag folded origami structures, at large plastic strains were investigated by numerical and analytical methods; it was found that the proposed origami metamaterials outperformed the conventional Miura-ori based metamaterials, in terms of energy absorption [29]. Wen and co-workers [30] proposed a class of novel origami metamaterials based on crease customization and stacking strategies; they numerically and analytically uncovered the tailored multistage stiffness. Zhou et al. [10] designed experiments to study the brace hysteretic behavior of a novel origami energy dissipation brace, formed by a combination of Miura and Tachi unit cells. Nevertheless, most of the aforementioned origami metamaterials are composed of open Miura origami, which obviously has a relatively poor out-of-plane stiffness [31], leading to limitations to engineering applications.

Combining two open Miura origami into a closed origami tube can not only improve the problem of poor out-of-plane stiffness, but also has excellent mechanical properties [32,33,34,35,36,37,38]. For instance, Liu and co-workers [33] showed that origami tubes can have a wide range of tunable dynamic properties, through experimental and numerical methods. Chen and co-workers proposed a theoretical design method for a family of origami tubes [37]. Origami tubes possessing polygonal cross-sections were designed using an analytical method, and their stiffness properties were investigated with physical tests [38]. Eidini et al. [39] seem to have been the first to study cellular folded mechanical metamaterials comprising different scales of zigzag origami strips. Subsequently, their work was extended to designing a class of one-degree of freedom cellular mechanical metamaterials [40]. Wang et al. [41] expanded the traditional Miura sheet and configured a variety of different cylindrical structures by combining two different crease patterns. By superimposing Miura sheets with different geometrical scales between multiple layers, Ma et al. [42] proposed a structure with gradient stiffness based on the basic Miura sheet. Experiments and numerical simulations verified that the structure had a better energy-absorbing capacity than the repeated layer structure with uniform unit cells.

Although closed origami structures have more outstanding mechanical properties than the open ones in some respects, scholars have normally concluded that the mechanical properties of the origami structure are mainly determined by its geometric topology, and that its constituent materials have a lesser influence. This conclusion has limitations in some situations [43,44]. One of the main reasons is that composite materials have a greater design freedom than single materials, which can greatly enrich the design possibilities of origami structures [45,46,47,48,49,50]. We can show a few representative works, as follows: Utilizing nonhomogeneous strain and folding techniques, novel 3D composite structures were achieved [46]. Carbon fiber reinforced plastic (CFRP) origami tubes were designed, and their excellent energy absorption capacity was identified [47]. Most recently, Du and co-workers developed a fabrication strategy inspired by origami techniques; they combined analytical, numerical, and experimental methods to investigate the mechanical properties of origami-inspired carbon fiber-reinforced composite sandwich materials [50]. In addition, it can be seen from the above literature review that most of the existing works are limited to quasi-static or static research, and there are few works on vibration characteristics [51,52]. It is well-known that vibration characteristics are a common concern in practical engineering, so it is extremely important to carry out research on this aspect of origami. This leads to the starting point of this work. Carbon fiber/epoxy resin composite materials have a higher strength, larger elastic modulus, low weight, and excellent chemical properties, and are comparable to metals in several applications. Gohari [53] and co-author conducted analytical, numerical, and experimental studies to improve the failure prediction of ellipsoidal domes laminate-woven using a CFRP composite material under internal pressure. Wang et al. [54] proposed an integrated modeling method for finite element analysis of the grinding process of long fiber-reinforced ceramic matrix woven composites and experimentally verified the effectiveness of the method. In another study by our group [43], we studied the relationship between the buckling load and design parameters of an origami metamaterial based on a classic Miura sheet and carbon fiber-reinforced composite materials. Considering these advantages, carbon fiber/epoxy resin was selected as the base material in this study.

Thus, inspired by the geometric topology design of the origami tube in [33,36,37,38], and making full use of the benefits of carbon fiber/epoxy resin composite materials, we studied the dynamic characteristics of composite material-based Miura-origami tube. We bonded two identical Miura sheets together to configure a tubular unit, which is referred to here as a Miura tube. Using numerical simulations, the relationships between the dynamic characteristics (including natural frequency and dynamic displacement response) of the carbon fiber/epoxy resin-based Miura tube and the structural parameters and material parameters were investigated, which provides a reference for further applications of the Miura tube structure. Notably, the Miura tube is regarded as a structure, instead of a mechanism. The reason for this can be found in [33]. We insist that the focus of our study is on the mechanical properties of a tubular structure with various folding states. Moreover, compared with metals, composite materials possess greater design freedom, since they can readily adjust their characteristics within a broad range. The organization structure of this paper is as follows: After the introduction, Section 2 describes how the Miura tube is constructed, including geometric modeling, finite element modeling, and settings of numerical simulations. Section 3 describes the simulation results and gives a discussion about the dynamic characteristics of the Miura tube, and Section 4 presents the conclusions.

## 2. Materials and Methods

### 2.1. Geometric Design of the Miura Tube

In this section, the classic Miura origami structure and the establishment of a Miura tube structure are described.

#### 2.1.1. Miura Sheet

As shown in Figure 1a, each Miura sheet unit consists of four equal parallelograms. The four parallelograms are uniquely determined by length *a*, width *b*, and acute angle *β*, and the geometric relationship between these parallelograms is determined by the intersection angle *θ*. Therefore, the geometric structure of a Miura sheet unit can be determined by four independent parameters: the parallelogram length *a*, width *b*, acute angle *β*, and the intersection angle *θ* between the parallelograms (or parallelogram length *a*, side length ratio *a*/*b*, acute angle *β*, and the angle between the parallelograms *θ*). In addition, the other parameters *w*, *l*, *v*, *h*, and side corners *γ* and *δ* shown in Figure 1a can be calculated using the following formula:$$cos\gamma =\frac{sin^{2}\beta cos^{2}\left(\theta /2\right)-cos^{2}\beta }{sin^{2}\beta cos^{2}\left(\theta /2\right)+cos^{2}\beta }\;cos\delta =sin^{2}\beta cos\theta -cos^{2}\beta \;w=2b\;sin\left(\gamma /2\right)\;h=a\;cos\left(\delta /2\right)\;l=2a\;sin\left(\delta /2\right)\;v=b\;sin\left(\gamma /2\right)$$

A classic Miura sheet model can be obtained by repeatedly arranging *n* Miura sheet units along the *x* and *y* directions. Notably, as shown in Figure 1b, the Miura sheet established has four Miura sheet units along the *x* direction, but only one along the *y* direction.

#### 2.1.2. Miura Tube

As shown in Figure 1c, the Miura tube described in this study can be obtained by stitching together two identical Miura sheets. In this situation, the geometric structure of the Miura tube is uniquely determined by five parameters: length *a*, width *b*, acute angle *β*, folding angle *θ*, and the number of arrangements of Miura sheet units *n*. Compared with the aforementioned Miura sheet, the tubular structure has a closed cross-section; hence, it exhibits better mechanical properties, especially the out-of-plane stiffness, which is also the focus of this research. Note that the geometrical design of this Miura tube is identical to that shown in [34]. However, differently, our research is primarily aimed at exploring the dynamic characteristics of a Miura tube made of carbon fiber/epoxy composite material.

### 2.2. Finite Element Modelling

Herein, a commercial nonlinear finite element code, Abaqus (Dassault Systemes Simulia Corp, Providence, RI, USA), was employed for numerical simulations [55], to obtain the natural frequency and other dynamic characteristics of the Miura tube. Miura sheets 1 and 2 are rigidly connected at their open sides. Herein, a Miura tube is considered a cantilever beam with one end fixed and the other end loaded. A four-node shell element (S4R) was used in meshing. The material parameters of the carbon fiber/epoxy composite material used in the numerical simulation are listed in Table 1 [56]. Notably, the arrangement number of Miura sheet element *n* in the model is 4, and the loaded end was applied with a force with sinusoidal periodic variation. The specific loading situation is presented in Figure 2, and the force can be expressed as:$$\{\begin{array}{c} F=2sin\left(2\pi \;t\right)N\\ 0\leq t\leq 5 \end{array}$$

## 3. Results and Discussion

### 3.1. The Natural Frequency

#### 3.1.1. Effects of Structural Parameters on the Natural Frequency (NF)

Herein, we employed a comparative research method to explore the relationship between the natural frequency of the Miura tube and its structural parameters. The variables set primarily included the wall thickness *t* of the Miura tube, the folding angle *θ*, and the side length ratio of the parallelogram *a*/*b*. Other structural parameters, such as the acute angle *β =* 55°, the side length of parallelogram *a =* 10 mm, and the number of Miura sheet units *n =* 4 remained unchanged. The numerical simulation results are listed in Table 2.

As shown in Table 2, when other variables remained unchanged, it can be clearly seen that the first three order natural frequencies of the Miura tube were gradually increased by increasing the wall thickness. The fundamental frequency reached a maximum value of 1080 Hz when *t* = 0.8 mm. By enlarging the side length ratio from 1 to 2.2, the first three order natural frequencies of the Miura tube were more obviously increased. The experimental group with *a*/*b* = 2.2 had the maximum fundamental frequency of 4705 Hz. When we fixed *a* = 10 mm, *a*/*b* = 1, *t* = 0.6 mm, *β* = 55°, and *ϕ* = 0° (*ϕ* is the fiber layup angle), the third order natural frequency increased gradually with the increase of the folding angle of the Miura tube, and the maximum fundamental frequency was 974 Hz at *θ* = 110°. To better present the change of natural frequency, we normalized the simulation data by the normalized natural frequency (NNF), which is shown in Figure 3. Again, it can be clearly seen that the NNF can be greatly altered by separately varying the thickness, the side length ratio, and the folding angle.

#### 3.1.2. Effects of Material Parameters on the Natural Frequency

Next, the relationship between the natural frequency of the Miura tube and the laying angle of the carbon fiber/epoxy resin composite material is explored. Notably, the structural parameters of the Miura tube were set as fixed values, where the side length of parallelogram *a* = 10 mm, the parallelogram side length ratio *a*/*b* = 1, the folding angle *θ* = 130°, the acute angle *β* = 55°, and the number of Miura sheet units *n* = 4. In addition, without a loss of generality, all the carbon fiber/epoxy sheets were composed of three layers, and the thickness of each layer was *t*_0_ = 0.2 mm, i.e., the thickness of the material sheet was 0.6 mm, and the layers were bonded. The laying Angle of carbon fiber in each layer in each scheme is shown in Figure 4. The numerical simulation results are presented in Table 3.

As per Table 3, when the laying angles of carbon fiber in the three-layer composite sheet were 90°, 90°, and 90°, respectively, the Miura tube demonstrated the lowest first three-order natural frequency, where the fundamental frequency was 647 Hz. When the laying angles of the carbon fiber were 0°, 90°, and 0°, the Miura tube displayed the highest fundamental frequency of 1052 Hz. It was also found that, by simply changing the ply angle, the natural frequencies of the first, second, and third orders can be adjusted within the ranges of 647–1052 Hz, 1027–1580 Hz, and 3608–5489 Hz, respectively. This also confirms the importance of using a carbon fiber/epoxy resin composite material to form origami tubes.

### 3.2. Study on Dynamic Displacement Response

The relationship between the dynamic displacement response (DDR) of a Miura tube and its structural and material parameters is further investigated in this section. The main variables set were the wall thickness *t* of the Miura tube, the folding angle *θ*, and the side length ratio of parallelogram *a*/*b*. Other structural parameters such as acute angle *β* = 55°, parallelogram side length *a* = 10 mm, and the number of Miura sheet units *n* = 4 remained constant.

#### 3.2.1. Effects of the Wall Thickness on the DDR

As depicted in Figure 5a, as the thickness of the Miura tube gradually decreased from 0.8 mm to 0.2 mm, the displacement response gradually became larger. When the thickness was 0.2 mm, the displacement response reached 0.0672 mm, while the displacement responses of the other three groups were all negligible, without any prominent differences. We can conclude that the origami tube has a high axial stiffness, since we considered the origami tube as a structure, which is consistent with what we claimed in the introduction. A similar phenomenon can be seen in Figure 5b, where the displacement response gradually became larger as the thickness of the Miura tube gradually became smaller. Specifically, when the thickness was 0.2 mm, the displacement response reached a maximum value of 0.8678 mm; this situation is sometimes not allowed in real engineering. The *xOz* out-of-plane stiffness is much larger than that of the *yOz*. In the third case, when the thickness was 0.2 mm, the displacement response reached as much as 2.8550 mm, as shown in Figure 5c, demonstrating that the *xOy* out-of-plane stiffness is significantly low.

#### 3.2.2. Effects of the folding angle *θ* on the DDR

As per Figure 6a, as the folding angle of the Miura tube increased, the DDR gradually decreased. When the folding angle was *θ* = 50°, the DDR reached a maximum value of 0.019 mm. As per Figure 6b, as the folding angle of the Miura tube increased, the displacement response did not show a uniform variation, but the displacement response reached 0.6617 mm when the folding angle was *θ* = 70°. Similarly, as per Figure 6c, as the folding angle of the Miura tube increased, the displacement response did not show a uniform variation law, but the displacement response reached maximum value of 0.3674 mm when the folding angle was *θ* = 110°. It was also found that the axial tensile stiffness and the *x**Oz* out-of-plane stiffness were more sensitive to the folding angle, while the *x**Oy* out-of-plane stiffness was smaller. We also revealed that the stiffness of the origami tube can be extensively tuned by merely altering the folding angle. Varying the folding angle can be readily achieved by rearranging the crease pattern [33,42]. Thus, this is an effective solution to changing the stiffness of the lightweight structure in a wide range.

#### 3.2.3. Effects of the Parallelogram Side Length Ratio *a*/*b* on the DDR

Figure 7 reveals that the side length ratio seems to have a greater impact on the stiffness of the origami tube than other factors, which can significantly affect the three types of stiffness. As per Figure 7a, as the side length ratio gradually increased, the displacement response did not show a uniform variation, but reached a maximum value of 0.00152 mm when *a*/*b* = 1 and the displacement response was minuscule (9.946 × 10^−5^ mm) when *a*/*b* = 1.8. Figure 7b shows that as *a*/*b* gradually increased, the displacement response gradually decreased. When *a*/*b* = 1, the displacement response reached a maximum value of 0.1708 mm. Figure 7c displays that as the *a*/*b* gradually increased, the displacement response gradually decreased. When *a*/*b* = 1, the displacement response reached a maximum value of 0.3763 mm. It was also found that the *x**Oy* out-of-plane stiffness was the smallest, compared with the other two, meaning that designers need to pay special attention to this stiffness when designing lightweight structures. Of course, this result also provides an effective method for designers to improve such stiffness.

### 3.3. Effects of the Number of Arrangements of Miura Sheet Units

The number of unit cells is also one of the most important structural parameters of an origami tube. To prove that could use four units to form an origami tube without loss of generality, additional numerical experiments were conducted. The main variables set were the number of arrangements of Miura sheet units *n*. Other structural parameters such as acute angle *β* = 55°, parallelogram side length *a* = 10 mm, the wall thickness of the Miura tube *t* = 0.6 mm, the folding angle *θ* = 130°, and the side length ratio of the parallelogram *a/b* = 1 remained constant.

As can be seen from Figure 8, when the number of the unit cells gradually increased from 2 to 10, with an interval of 2, the NNFs of the designed origami tube gradually decreased and converged approximately to a constant value. In particularly, when *n* = 4, the first three normalized frequencies became nearly equal. As can be seen from Figure 9, the dynamic response displacement in the three directions of the Miura tube increased with the increase of the unit cells. This can be easily understood, in that lengthening the structure will reduce the stiffness of the structure, especially for the two types of out-of-plane stiffness.

## 4. Conclusions

A novel Miura tube with closed sections was obtained by bonding two identical Miura sheets, with the aid of origami techniques and the good material parameters of the carbon fiber/epoxy resin. The dynamic characteristics of the carbon fiber/epoxy resin-based Miura tube were investigated through numerical simulations. By adjusting the material characteristics and structural parameters, such as the wall thickness *t* of Miura tube, the parallelogram side length ratio *a*/*b*, and the folding angle *θ*, the dynamic characteristics could be significantly changed within a more extensive range. Thus, by altering the structural parameters and material parameters, one can effectively reduce the DDR and enlarge the NF. This provides effective methods for designing lightweight structures with high stiffness and the desired natural frequency. The proposed Miura tube in the study has several benefits; for instance, it can be (1) applied to various engineering fields, such as the core layer of a sandwich material, energy-absorbing structures [9,10,11], and so on; (2) fabricated from inexpensive two-dimensional metal materials but possess outstanding dynamic properties [33,34]; and (3) used as the initial design domain of a topology optimization to further enhanced the desired mechanical properties [57,58,59,60,61]. However, we still have some works in progress, for instance, fabricating a carbon fiber/epoxy resin-based origami tube [62,63,64], conducting modal tests [33], and performing a structural optimization to further enhance the dynamic properties of the origami tube [57,58,59,60,61]. These results will be published in another paper.

## Figures and Tables

**Figure 1 materials-14-06374-f001:**
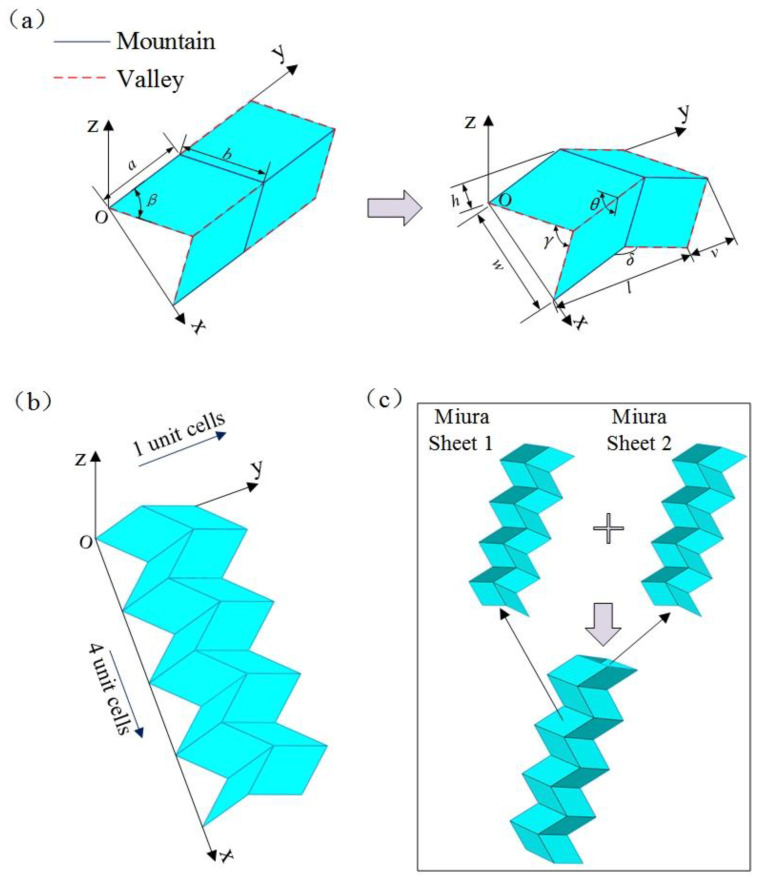
Construction of a Miura tube model: (**a**) geometric model of Miura sheet unit, (**b**) geometric model of a Miura sheet (consisting of four Miura sheet units), and (**c**) geometrical model of a Miura tube (consisting of two identical Miura sheets).

**Figure 2 materials-14-06374-f002:**
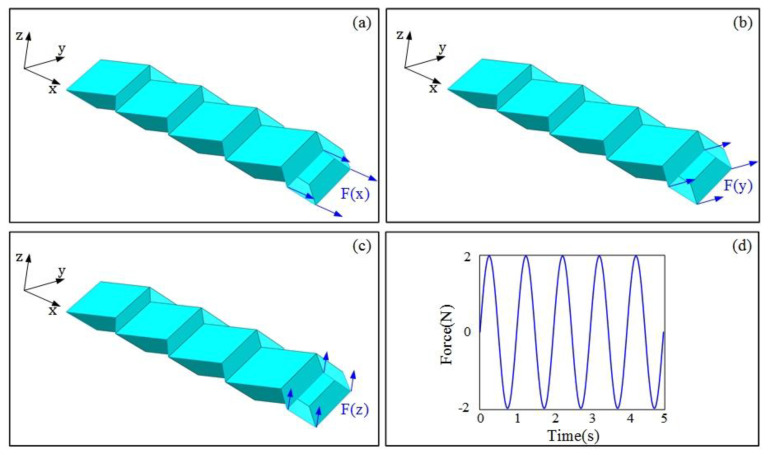
(**a**–**c**) illustrate the concentrated forces along the *x*, *y*, and *z* directions, respectively, acting on the four nodes at the free end. (**d**) The variation of periodic force.

**Figure 3 materials-14-06374-f003:**
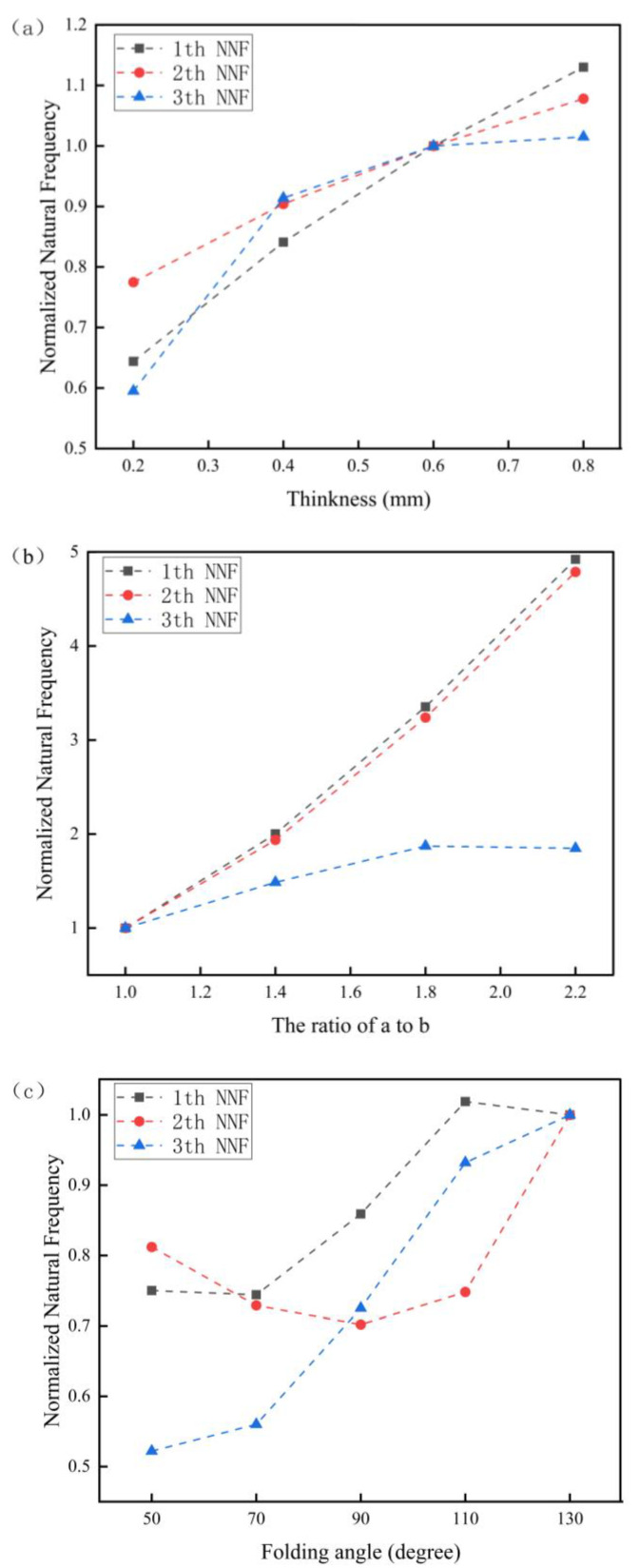
Numerical simulation results of the NF of the Miura tube. (**a**) NNF variation with wall thickness of the Miura tube (**b**). NNF variation with side length ratio of the Miura tube. (**c**) NNF variation with folding angle of the Miura tube.

**Figure 4 materials-14-06374-f004:**
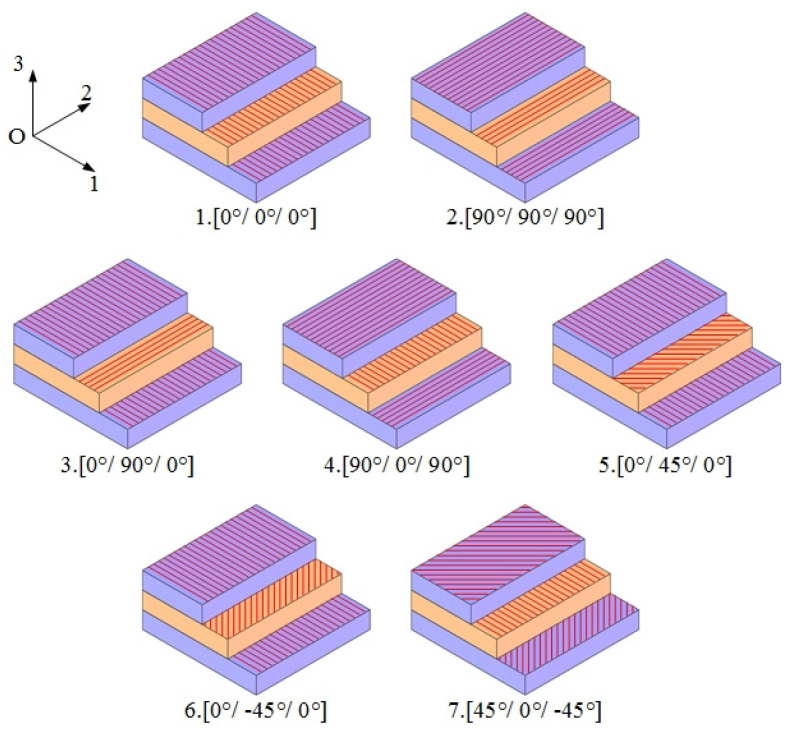
Layout scheme of each carbon fiber layer.

**Figure 5 materials-14-06374-f005:**
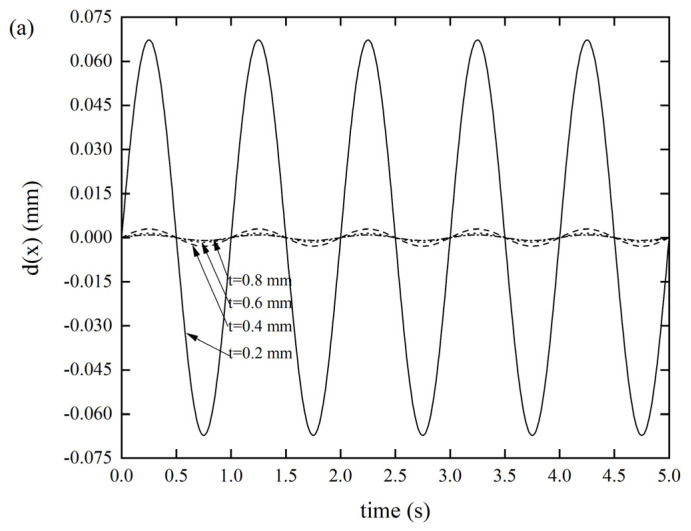
DDR of the Miura tube with different thicknesses *t*(*θ* = 130°, *a/b* = 1) (**a**) *x*-displacement of the Miura tube under force along the *x*-direction, (**b**) *y*-displacement of the Miura tube under force along the *y*-direction, and (**c**) *z*-displacement of the Miura tube under force along the *z*-direction.

**Figure 6 materials-14-06374-f006:**
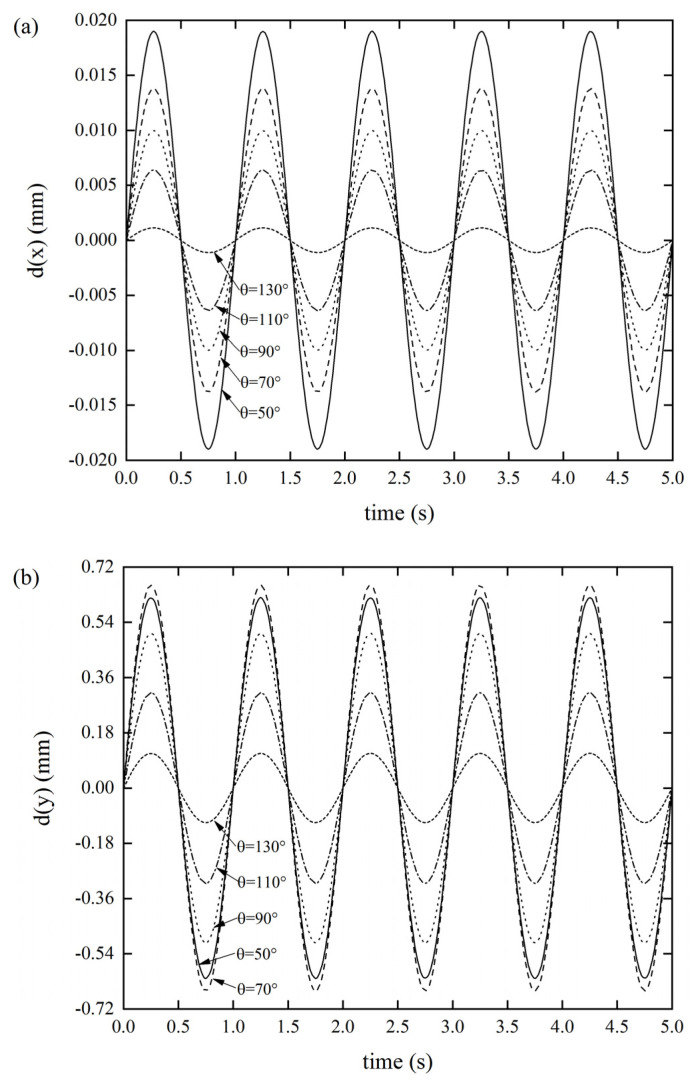
DDR of the Miura tube when folding angle *θ* varies(*t* = 0.6 mm, *a/b* = 1) (**a**) *x*-displacement of the Miura tube under force along the *x*-direction, (**b**) *y*-displacement of the Miura tube under force along the *y*-direction, and (**c**) *z*-displacement of the Miura tube under force along the *z*-direction.

**Figure 7 materials-14-06374-f007:**
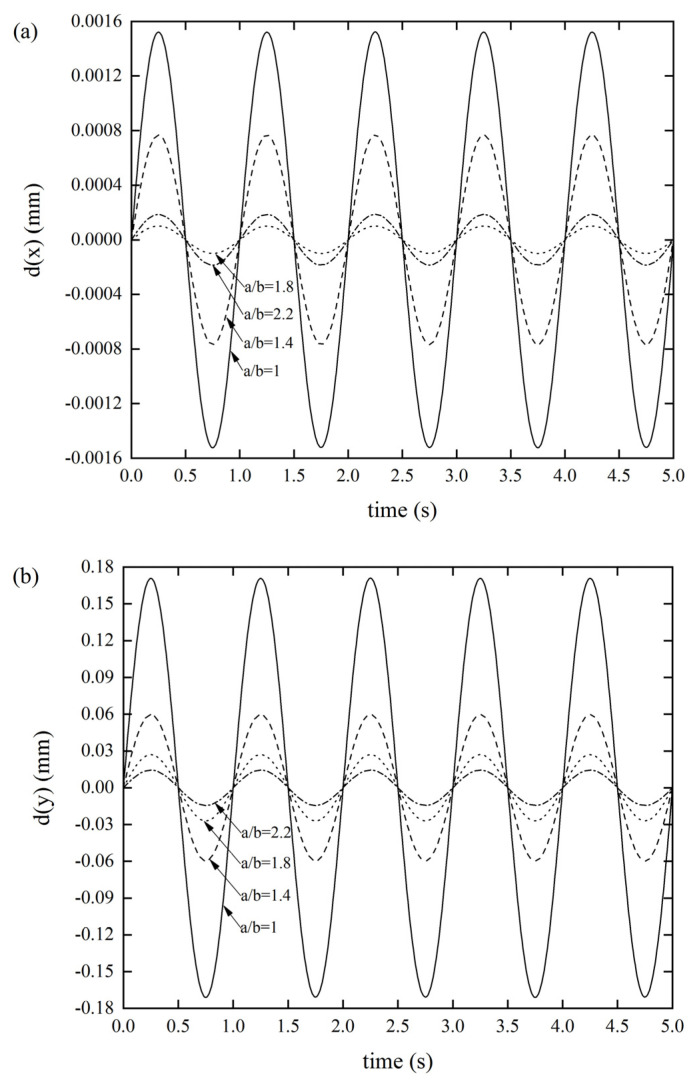
DDR of the Miura tube for different side length ratios of parallelogram *a*/*b*(*θ* = 130°, *t* = 0.6 mm) (**a**) *x*-displacement of the Miura tube under force along the *x*-direction, (**b**) *y*-displacement of the Miura tube under force along the *y*-direction, and (**c**) *z*-displacement of the Miura tube under force along the *z*-direction.

**Figure 8 materials-14-06374-f008:**
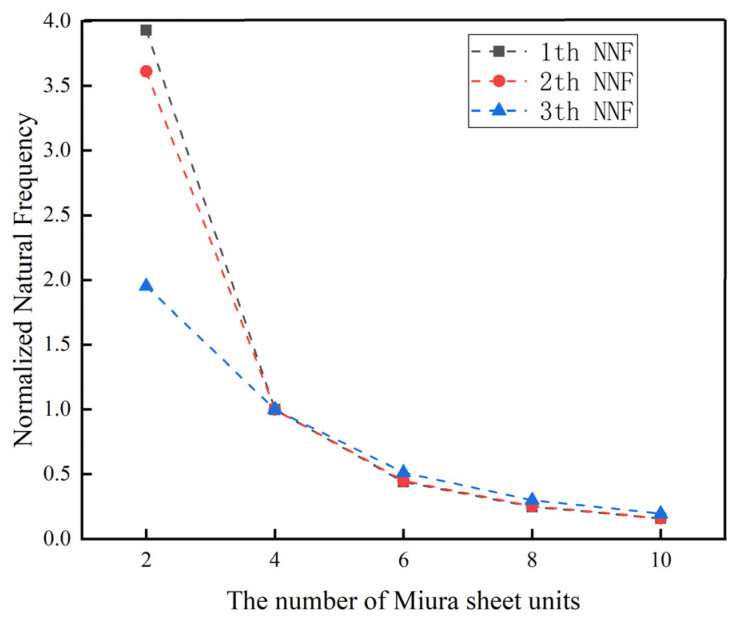
Numerical simulation results of the NNF of the Miura tube when the number of arrangements of Miura sheet units varied.

**Figure 9 materials-14-06374-f009:**
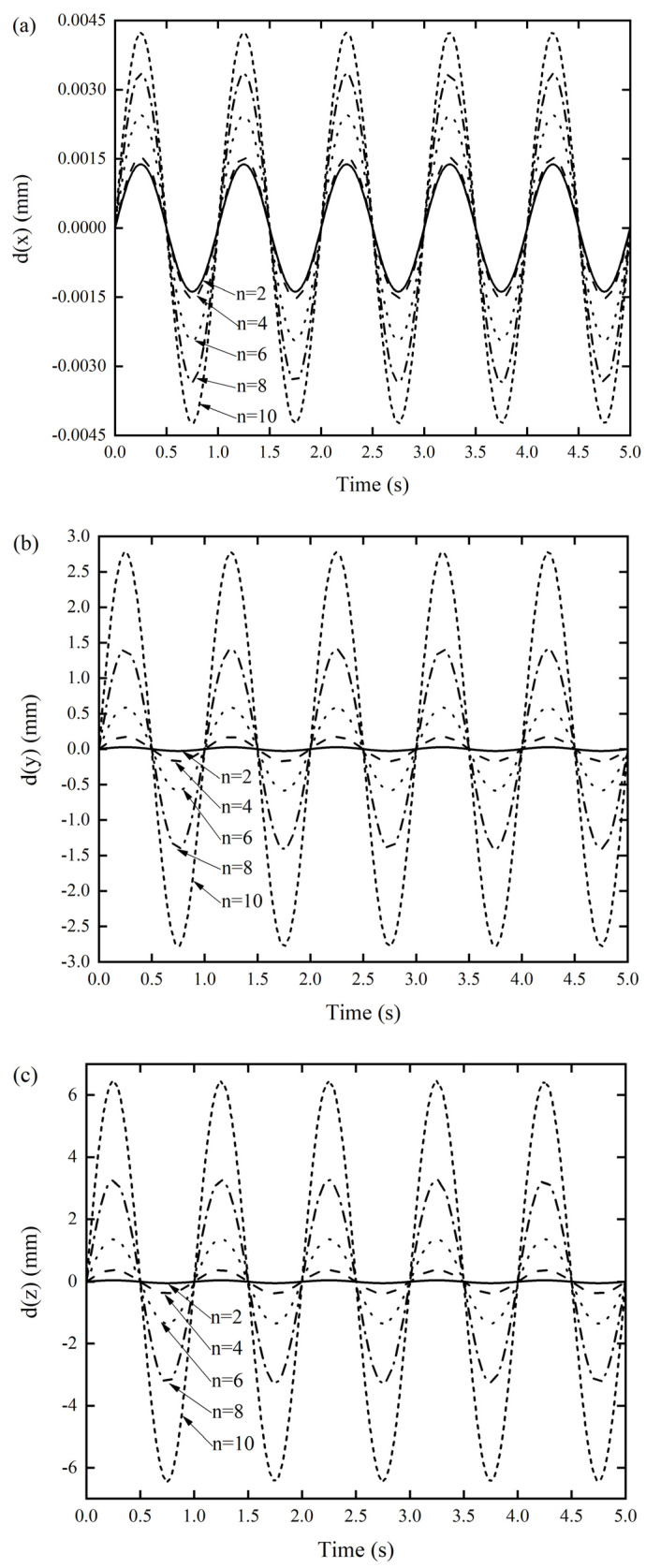
DDR of the Miura tube when the number of arrangements of Miura sheet units varied (**a**) *x*-displacement of the Miura tube under force along the *x*-direction, (**b**) *y*-displacement of the Miura tube under force along the *y*-direction, and (**c**) *z*-displacement of the Miura tube under force along the *z*-direction.

**Table 1 materials-14-06374-t001:** Material characteristics of the carbon fiber/epoxy resin composite material.

Material Properties	Values	
Young’s modulus/GPa	*E* _1_	121
	*E* _2_	8.6
	*E* _3_	8.6
Shear modulus/GPa	*G* _12_	4.7
	*G* _13_	4.7
	*G* _23_	3.1
Poisson’s ratio	*v*	0.27
density/(kg·m^−3^)	*ρ*	1490

**Table 2 materials-14-06374-t002:** Numerical simulation results of NF of the Miura tube.

Fixed Parameters	Changeable Parameters	NF		
		ω_1_/Hz	ω_2_/Hz	ω_3_/Hz
*a* = 10 mm, *a*/*b* = 1 *θ* = 130°, *β* = 55° *ϕ* = 0°	*t* = 0.2 mm	616	1084	2785
	*t* = 0.4 mm	804	1264	4277
	*t* = 0.6 mm	956	1398	4677
	*t* = 0.8 mm	1080	1507	4745
*a* = 10 mm, *t* = 0.6 mm *θ* = 130°, *β* = 55° *ϕ* = 0°	*a*/*b* = 1	956	1398	4677
	*a*/*b* = 1.4	1914	2709	6956
	*a*/*b* = 1.8	3205	4528	8759
	*a*/*b* = 2.2	4705	6695	8647
*a* = 10 mm, *a*/*b* = 1 *t* = 0.6 mm, *β* = 55° *ϕ* = 0°	*θ* = 50°	717	1135	2443
	*θ* = 70°	712	1019	2804
	*θ* = 90°	821	982	3390
	*θ* = 110°	974	1046	4359
	*θ* = 130°	956	1398	4677

**Table 3 materials-14-06374-t003:** NNF simulation results of a multi-layered Miura tube.

Group	Layout Scheme	NNF		
		*ω*_1_/Hz	*ω*_2_/Hz	*ω*_3_/Hz
1	0°/0°/0°	956	1398	4677
2	90°/90°/90°	647	1027	3608
3	0°/90°/0°	1052	1572	5134
4	90°/0°/90°	899	1473	4766
5	0°/45°/0°	1006	1580	5489
6	0°/−45°/0°	1002	1539	5315
7	45°/0°/−45°	728	1322	4008

## Data Availability

The data and results involved in this study have been presented in detail in the paper.

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
