# Peer review of "Revealing the Dynamic Characteristics of Composite Material-Based Miura-Origami Tube"

_materials, 2021, doi:10.3390/ma14216374_

Round 1

Reviewer 1 Report

SUMMARY AND RECOMMENDATION

This manuscript reports numerical investigations of the dynamic response of Miura-fold cantilever beam constructed of a carbon fiber/epoxy composite. The novelty of the results seems to hinge on the reporting of natural frequencies and harmonic response as opposed to the folding mechanics presented in previous studies; the stated value of the work is thus for informing the design of structures using Miura tubes. These points, however, are not well-motived and not supported by the results. The authors do not make a case for the results being relevant to systems beyond the one presented here. The writing is poor. I believe the manuscript requires SUBSTANTIAL revision in order to prove value to the journal readers. I cannot recommend this article for publication in its present state.

MAJOR CONCERNS

Page 1, Lines 29-33: The authors make a number of claims without citation. For example, the authors should cite a paper or two Miura who introduced the concept. The authors also claim Miura structures have unique mechanical properties without giving any examples and without citation.

Page 2, Line 87: The authors say that they are inspired by Ref. [29], but there has been no but it is not clear how, leaving the reader to wonder what influence does Ref. [29] have on the authors' research.

In general, the literature review is not well written. The authors cite several references but do not relate these references to the work done in this manuscript. As such, it seems as if the authors are simply citing any articles on Miura structures whether they are relevant or not. The reviewer would very much appreciated if the authors could edit the introduction to more effectively tie cited article together and relate them more strongly to the authors' research.

Trends from the parameter study are not sufficiently analyzed: no insight is provide and no formulaic relations are suggested. The authors claim that their results will influence future design, but without insight and new formulae, there is nothing for the reader to take away or build off of.

All data reported is for 4 units of the Miura tube, and results are sure to vary with this key parameter. Seeing as the research is intended to benefit design, omission of this key parameter is a critical flaw.

Section 3.2 reports the dynamic response of the miura tube under loading specified in section 2.2. These results in my opinion lack value for designers, as the natural frequency of the beam is enough to inform design. Response of the beam to harmonic loading seems redundant in this case.

What new and interesting behavior result from using carbon fiber layers that can’t be achieved otherwise?

To inform design using any material, I would suggest non-dimensionalizing parameters. This would allow the design of structures using any material consisting of a substrate with embedded fibers.

MEDIUM CONCERNS

Page 4, Line 144--145: The sentence beginning "For the calculations of..." is confusion. Perhaps two sentences have accidentally been merged into one. Regardless, the authors should consider re-phrasing.

Page 5, Line 162: The authors mention a "normalized natural frequency (NNF)" but do not define it. What is the frequency normalized by? If the frequency is normalized, then it should be dimensionless, but all the figures with NNF are in units of Hertz (Hz) which has units of inverse seconds. This is contradictory and should be clarified or corrected.

There are two Figure 3’s in the manuscript. There are three Figure 4's in the manuscript. Also, the are multiple instances, where the author present information in figures, and then present the same information in tables…this is not concise.

On page 2, the authors assert “Carbon fiber/epoxy resin composites have higher strength…” with no citation of reference. A similar assertion follows at the bottom of page 2.

Motivation of the project is convoluted. The introduction offers a catalog of research on Miura fold structures, but does not effectively link them together to motivate authors’. Some clue is given on page 3 in section 3.2: “Different from previous literature…” but this is far into the manuscript.

MINOR CONCERNS

Page 3, Lines 112-115: The sentence beginning with "In other words,..." is completely superfluous; it does not provide any new information.

Multiple spelling and grammatical errors, largely on the use of plurals. There are also multiple spellings of Miura (sometimes muira).

The authors reference “the comparative research method” on page 7 which seems to be an unnecessarily elaborate way of describing a “parameter study”.

Author Response

The authors would like to express their great gratitude to the reviewers for their constructive comments and recognition of the technical soundness of the proposed methods. We have carefully considered all the issues and incorporated them into the revisions in line with these invaluable comments raised.See the attachment for specific modification information

Reviewer 2 Report

Comments on “Revealing the Dynamic Characteristics of Composite Material-
based Miura-Origami Tube”, by Houyao Zhu, Zhixin Li, Ruikun Wang, Shouyan Chen, Chunliang Zhang and Fangyi Li

The Authors present numerical results for mechanical properties (relationship between normalized natural frequency (NNF) of the Miura tube and its structural parameters) of a newly obtained Miura tube embedded into an epoxi-resin. The Miura tube is obtained by bonding “two identical Miura sheets together to configure a tubular unit” giving what is called a Miura tube. Regarding the
numerical simulation part, the Authors used Abaqus commercial software. “Miura tube was considered as a cantilever beam with one end fixed and the other end loaded”. The study is intended to be exploratory in nature, no comparisons with experimental data are performed. The various numerical results are gathered and discussed in Section 3. However, even though the paper is uniquely numerical simulations oriented, one does not see any technical detail regarding the assumptions laying underneath the numerical work. They are necessary in order to make a better assessment of the value of the simulations.

The paper is certainly very interesting and useful to those acting in the area of advanced composite materials, but additions are needed, and I am willing to recommend its publications upon revision.

Author Response

(The authors gave the same response as above.)

Round 2

Reviewer 2 Report

I have no new comments at this stage.

Author Response

Reviewer 2' comments: Origami nano-composites, the absence of governing equations (and the corresponding lack of details) is a quite noticeable weakness. For instance, the Authors address the important problem of “Effects of the number of arrangements of Miura sheet units” and give numerical results, but give no indication as to how those arrangements are modeled. Is this a confidential matter?

Reply: Thank you very much for your comments and suggestions, which will help us better improve our paper and also provide great guidance for our subsequent research work. Indeed, governing equations allow us to establish more specific and universal laws or theories between model parameters and numerical simulation results, which is more conducive to predicting structural characteristics.This is clearly more exciting research. However, in the research of this paper, we are inspired by traditional origami structures to design some new metamaterials and explore their properties. The structures of these metamaterials are often complex, and their theoretical models or governing equations are difficult to obtain.For this reason, we give priority to using numerical simulation methods for preliminary research, hoping that these research conclusions can provide reference for other designers. To be sure, we will continue to pay attention to the theoretical construction method of the governing equation of this structure in the subsequent research. In addition, as for some details, we are really sorry for our negligence, and we have made modifications. We added some details about 3D modeling parameters in the second revision. Thank you again for your comments and suggestions, and we hope that our answers can be approved by you.
